# INCORPORATING EXPLICIT UNCERTAINTY ESTIMATES INTO DEEP OFFLINE REINFORCEMENT LEARNING

## ABSTRACT

Most theoretically motivated work in the offline reinforcement learning setting requires precise uncertainty estimates. This requirement restricts the algorithms derived in that work to the tabular and linear settings where such estimates exist. In this work, we develop a novel method for incorporating scalable uncertainty estimates into an offline reinforcement learning algorithm called deep-SPIBB that extends the SPIBB family of algorithms to environments with larger state and action spaces. We use recent innovations in uncertainty estimation from the deep learning community to get more scalable uncertainty estimates to plug into deep-SPIBB. While these uncertainty estimates do not allow for the same theoretical guarantees as in the tabular case, we argue that the SPIBB mechanism for incorporating uncertainty is more robust and flexible than pessimistic approaches that incorporate the uncertainty as a value function penalty. We bear this out empirically, showing that deep-SPIBB outperforms pessimism based approaches with access to the same uncertainty estimates and performs at least on par with a variety of other strong baselines across several environments and datasets.

## 1 INTRODUCTION

In the study of offline reinforcement learning (OffRL), uncertainty plays a key role (Buckman et al., 2020; Levine et al., 2020). This is because, unlike online RL where an agent receives feedback in the form of low rewards after taking a bad action, an OffRL agent must learn from a fixed dataset without feedback from the environment. As a result, a consistent issue for OffRL algorithms is the overestimation of states and actions that are not seen in the dataset, leading to poor performance when the agent is deployed and finds that those states and actions in fact have low reward (Fujimoto et al., 2019b). To overcome this issue, OffRL algorithms often attempt to incorporate some notion of uncertainty to ensure that the learned policy avoids regions of high uncertainty.

There are two main issues with this approach: (1) how to define uncertainty and (2) how to incorporate uncertainty estimates into the OffRL algorithm. In tabular and linear MDPs, issue (1) is resolved by using visitation counts and elliptical confidence regions, respectively (Yin et al., 2021; Yin & Wang, 2021; Jin et al., 2021; Laroche et al., 2019). In the large-scale MDPs that we consider, neither of these solutions work, but there is a large literature from the deep learning community on uncertainty quantification that we can leverage for OffRL (Ciosek et al., 2019; Osband et al., 2018; 2021; Burda et al., 2019; Ostrovski et al., 2017; Lakshminarayanan et al., 2017; Blundell et al., 2015; Gal & Ghahramani, 2016). Given these uncertainty estimators, this paper focuses primarily on issue (2), how to incorporate uncertainty for OffRL.

To understand how to best incorporate uncertainty into an OffRL algorithm, we first provide a high level algorithmic template that captures the majority of related work as instances of modified policy iteration (Scherrer et al., 2012) that alternate between policy evaluation and policy improvement. We then can sort prior work into four categories along two axes: whether the algorithm modifies the evaluation step or the improvement step, and whether the algorithm uses an explicit uncertainty estimator or not. One class of algorithms modifies the evaluation step by introducing value penalties based on explicit uncertainty estimates, which we will call pessimism (Petrik et al., 2016; Buckman et al., 2020; Jin et al., 2021). An alternative modifies the value estimation without using an uncertainty estimate, like in CQL (Kumar et al., 2020). Another family uses behavior constraints that modify the policy improvement step to keep the learned policy near the behavior policy (Fujimoto et al.,

2019b;a), but does not use explicit uncertainty. Instead, we propose to use the fourth class of methods that leverages uncertainty-based constraints in the policy improvement step and is inspired by the SPIBB family of algorithms (Laroche et al., 2019; Laroche & Tachet des Combes, 2019; Nadjahi et al., 2019; Simão et al., 2020). These algorithms modify the policy improvement step like a behavior constraint, but also reason about state-based uncertainty like the pessimistic algorithms. Explicitly, we define the deep-SPIBB algorithm that effectively incorporates uncertainty estimates into OffRL.

The main contributions of this paper are as follows:

- We introduce the deep-SPIBB algorithm which provides a principled way to incorporate scalable uncertainty estimates for OffRL. We instantiate this algorithm using ensemble-based uncertainty estimates inspired by Bayesian inference (Ciosek et al., 2019; Osband et al., 2021).

- We provide a detailed comparison of several different mechanisms to incorporate uncertainty by considering how each mechanism operates at the extreme settings of its hyperparameters. This analysis shows that deep-SPIBB provides a flexible and robust mechanism to interpolate between various extremes (greedy RL, behavior cloning, and one-step RL).

- Through experiments on classical environments (cartpole and catch) as well at Atari games, we demonstrate the efficacy of deep-SPIBB. In particular, we find that deep-SPIBB consistently outperforms pessimism when given access to the same imperfect uncertainty estimators.

- When deep-SPIBB has access to better uncertainty estimators (as in the easier cartpole environment) it is able to substantially outperform our other baselines of CQL and BCQ as well. This suggests that as uncertainty estimators improve, deep-SPIBB will provide a useful mechanism for incorporating them for OffRL.

## 2 PRELIMINARIES

We consider an OffRL setup with a discrete action space and access to a dataset $\mathcal{D} = \{(s_j, a_j, r_j, s'_j)\}_{j=1}^N$ consisting of $N$ transitions collected by some behavior policy $\beta$. The goal is to learn a policy $\pi$ from this data to maximize expected discounted returns $J(\pi) = \mathbb{E}_{\tau \sim \pi}[\sum_{t=0}^{\infty} \gamma^t r_t]$.

### 2.1 ALGORITHMIC TEMPLATE

The vast majority of prior work on the OffRL problem can be seen through a common algorithmic template of modified policy iteration. Each algorithm alternates between policy evaluation and policy improvement steps. The main difference between algorithms comes in how they modify either the evaluation or the improvement step. Below we first define the generic version of the OffRL algorithmic template and then explain how different OffRL algorithms modify this template.

Policy improvement by greedy maximization:

$$\pi^{(i+1)}(\cdot|s) = \arg\max_{\pi \in \Pi} \sum_{a \in \mathcal{A}} \pi(a|s)\hat{Q}^{(i)}(s, a). \tag{1}$$

Value estimation by fitted Q evaluation given the dataset $\mathcal{D} = \{(s_j, a_j, r_j, s'_j)\}_{j=1}^N$. Define the Bellman operator from datapoint $j$ with $\pi^{(i+1)}, \hat{Q}^{(i)}$, as $\mathcal{T}(j, \pi, Q) = r_j + \gamma \sum_{a' \in \mathcal{A}} \pi(a'|s'_j)Q(s'_j, a')$. Then the evaluation step is:

$$\hat{Q}^{(i+1)} = \arg\min_{Q \in \mathcal{Q}} \sum_j \left( Q(s_j, a_j) - \mathcal{T}(j, \pi^{(i+1)}, \hat{Q}^{(i)}) \right)^2 \tag{2}$$

In addition to the policy and Q function, some algorithms we consider will also learn an estimated behavior policy $\hat{\beta}(a|s)$ and/or an uncertainty function $\hat{u}(s, a)$. Generally, $\hat{\beta}$ is learned by maximum likelihood supervised learning. The uncertainty $\hat{u}$ on the other hand can be learned many different ways. We will discuss $\hat{u}$ in more detail in Section 3 when we describe our method.

With this template we can provide a characterization of much prior work that is summarized in Table 1. The essential axes that we consider are (1) whether the algorithm modifies the improvement step or the evaluation step and (2) whether the algorithm uses an uncertainty function $u(s, a)$ or not.

Table 1: A characterization of how several baseline methods fit into our template.

|  | Uncertainty-free | Uncertainty-based |
|---|---|---|
| Modified improvement step | BCQ | deep-SPIBB (ours) |
| Modified evaluation step | CQL | Pessimism |

## 2.2 Modifying the evaluation step

One approach to incorporating uncertainty is to introduce a penalty into the value estimation step that encourages the policy to avoid novel states or actions.

**Uncertainty-based.** The simplest value penalty is to use an explicit estimate of uncertainty, we will can this algorithm *pessimism*. Variants of pessimism have been examined in a broad range of prior work (Petrik et al., 2016; Buckman et al., 2020; Jin et al., 2021). Given an uncertainty estimator $u : \mathcal{S} \times \mathcal{A} \to \mathbb{R}$, the pessimism algorithm modifies the evaluation step to be:

$$\hat{Q}^{(i+1)} = \underset{Q \in \mathcal{Q}}{\arg\min} \sum_j \left( Q(s_j, a_j) - \left( \mathcal{T}(j, \pi^{(i+1)}, \hat{Q}^{(i)}) - \alpha \cdot u(s_j, a_j) \right) \right)^2 \qquad (3)$$

The hyperparameter $\alpha$ controls the amount of pessimism.[1]

**Uncertainty-free.** Alternatively, the algorithm can modify the evaluation step without use of an explicit uncertainty function. A popular representation of this approach is the CQL algorithm Kumar et al. (2020). In CQL there is no explicit estimate of uncertainty. Instead, the algorithm makes the following update in the evaluation step:

$$Q^{(i+1)} = \underset{Q \in \mathcal{Q}}{\arg\min} \sum_j \alpha \big( \log \sum_a \exp(Q(s_j, a)) - Q(s_j, a_j) \big) + \big( Q(s_j, a_j) - \mathcal{T}(j, \pi^{(i+1)}, \hat{Q}^{(i)}) \big)^2$$

$$(4)$$

The first term encourages the Q estimates to underestimate Q values at unobserved actions (via the log-sum-exp term) while remaining large at observed actions (via the $Q(s_j, a_j)$ term). Again the hyperparameter $\alpha$ controls the penalty.

As explained in the original paper, this version of CQL can be viewed as a version of pessimism with an entropy regularization term. In analog to pessimism, the implicit uncertainty function would take the form of $u(s, a) = \frac{\pi(a|s)}{\beta(a|s)} - 1$ where $\pi$ is the current policy iterate. Note this function is non-stationary since it depends on $\pi$. Moreover, in practice with neural function approximation, the CQL objective may behave differently than trying to implement this function explicitly. The implicit nature of the uncertainty used by CQL makes it different from standard pessimism with explicit uncertainty estimates.

## 2.3 Modifying the improvement step

Instead of modifying the evaluation step, we can alternatively modify the improvement step.

**Uncertainty-free.** The simplest way to modify the improvement step without using an uncertainty estimate is to constrain the learned policy to choose actions that are well-supported under the estimated behavior. The main example of this algorithm that we consider is the BCQ algorithm (Fujimoto et al., 2019b;a). Explicitly, the BCQ algorithm with hyperparameter $\tau$ defines

$$\pi^{(i+1)}(a|s) = \mathbb{1}\left[ a = \underset{a' \in \mathcal{A}_\tau(s)}{\arg\max} \hat{Q}^{(i)}(s, a') \right], \quad \mathcal{A}_\tau(s) = \left\{ a \in \mathcal{A} : \frac{\hat{\beta}(a|s)}{\max_{a' \in \mathcal{A}} \hat{\beta}(a'|s)} \geq \tau \right\} \quad (5)$$

where $\hat{\beta}$ is a maximum likelihood estimate of the behavior policy. When $\tau = 1$ this is exactly behavior cloning and when $\tau = 0$ the constraint has no effect. Importantly, these methods do not use any notion of uncertainty over states.

---

[1]Another instance of an uncertainty-based modification of the evaluation step is from the MBS algorithm of Liu et al. (2020). Instead of using an uncertainty penalty, they threshold an uncertainty function and propagate the minimal possible return in the Bellman backup if the uncertainty is too high.

**Uncertainty-based.** The final option is to modify the policy improvement step with the use of an explicit estimate of uncertainty. Our propsed deep-SPIBB algorithm falls into this category. In contrast to behavior constraints, an uncertainty-based constraint takes into account the confidence that we have in a given state. And in contrast to a value penalty, the uncertainty is not propagated in the evaluation step. The main example of this style of algorithm is the SPIBB family of algorithms (Laroche et al., 2019; Nadjahi et al., 2019; Simão et al., 2020) which we will discuss further in Section 3 when we introduce deep-SPIBB.

### 2.4 (Un)related work

In this subsection, we promptly acknowledge the existence of algorithm approaches for Offline RL that are not directly related with the uncertainty question that this work is endeavoring to address. A wide range of algorithms rely on actor-critic algorithmic architecture in order to handle MDPs with continuous actions Wang et al. (2020); Wu et al. (2019); Siegel et al. (2020); Kostrikov et al. (2021); Fujimoto & Gu (2021). We focus exclusively on the discrete action case and study different ways of incorporating uncertainty into the algorithm. Another group of algorithms take advantage of an explicit MDP model $\hat{m}$, which confers them better out-of-distribution generalization capabilities Yu et al. (2020); Kidambi et al. (2020); Yu et al. (2021); Janner et al. (2021). In our study, all the considered algorithms are model-free in order to guard ourselves against confounding factors of additional implementation details. Finally, there is the return-condition supervised learning approach that has recently been introduced in the Offline RL literature Chen et al. (2021); Emmons et al. (2021). We found the structure of these algorithms to be too distant from our work.

## 3 Deep-SPIBB

We can now explicitly define the deep-SPIBB algorithm. To make the algorithm scale up to high dimensional inputs, we use a neural uncertainty estimator described in Section 3.1. Using this uncertainty estimator, deep-SPIBB uses a variant of the approximate soft-SPIBB (Nadjahi et al., 2019) mechanism to incorporate uncertainty estimates into the policy improvement step, described in Section 3.2.

### 3.1 Neural uncertainty estimation

As presented in prior work, soft-SPIBB relies on count-based uncertainty estimates $\hat{u}(s, a)$ that have high probability guarantees. In the tabular case, we can derive these estimates from visitation counts $n(s, a)$ and set $\hat{u}(s, a) = c/\sqrt{n(s, a)}$. Unfortunately, this is not a scalable solution in larger domains since it is unknown how to best generalize these precise notions of uncertainty to larger domains that require data-efficient generalization across states. So, we borrow from the literature on neural uncertainty estimation (Ciosek et al., 2019; Osband et al., 2021) and use an uncertainty estimator based on ensembles trained to estimate random targets and regularized by random priors. The objective and estimator are described in full detail in Appendix A. At a high level, the uncertainty is proportional to the variance of an ensemble of models, each trained to predict a random function with high-dimensional outputs.

We make one key change relative to prior work. Rather than using both state and action as input to our uncertainty estimator, we only use the ensemble-based uncertainty estimator to estimate state-based uncertainty $\hat{u}(s)$. We then combine this with the behavior policy estimate $\hat{\beta}(a|s)$ to derive $\hat{u}(s, a)$. In particular, we define our uncertainty estimator as

$$\hat{u}(s, a) = \frac{\hat{u}(s)}{\sqrt{\hat{\beta}(a|s)}}. \tag{6}$$

The rationale for using $\sqrt{\hat{\beta}(a|s)}$ comes from the tabular setting, where if we were to use counts to define $\hat{u}(s) = \frac{c}{\sqrt{n(s)}}$ and $\hat{\beta}(a|s) = \frac{n(s,a)}{n(s)}$, then our $\hat{u}(s, a)$ would be exactly the standard count-based $\frac{c}{\sqrt{n(s,a)}}$. This decision is supported empirically in the experiments section. Intuitively, this helps because it guarantees that the uncertainty estimator is consistent with the estimated behavior used in the SPIBB constraint.

## 3.2 Algorithm

The algorithm consists of three models: $\hat{\beta}(a|s)$ an estimate of the behavior policy, $\hat{u}(s,a)$ an uncertainty quantification, and $\hat{Q}$ an estimated Q function. The Q updates are much like those of SPIBB-DQN (Laroche & Tachet des Combes, 2019) except we use approximate soft-SPIBB and scalable neural uncertainty estimates. Since we use target networks to approximate policy iteration, we will use parenthetical superscripts to keep track of the policy iteration step (i.e. the number of times the target network has been updated).

**Policy improvement step.** At step $i+1$, the policy $\pi^{(i+1)}$ approximates the solution to the following constrained optimization problem:

$$\pi(\cdot|s) = \arg\max_{\pi \in \Delta^{|\mathcal{A}|}} \sum_{a \in \mathcal{A}} \hat{Q}^{(i)}(s,a)\pi(a|s), \quad s.t. \quad \sum_{a \in \mathcal{A}} \hat{u}(s,a)\left|\pi(a|s) - \hat{\beta}(a|s)\right| \leq \epsilon_{train} \quad (7)$$

Since this problem is difficult to solve exactly, we use the approximation technique described in Nadjahi et al. (2019).

**Value estimation step.** For the value estimation step, we use the standard expected SARSA backup from the Equation (2). Since $\pi^{(i+1)}$ is uncertainty-constrained, it prevents Q values from being propagated from state-action pairs with high uncertainty under $\hat{u}$.

**Evaluation policy.** One modification that we make from prior work on soft-SPIBB is to generalize the algorithm by separating the hyperparameter $\epsilon_{train}$ that governs the deviation from the behavior during the policy improvement step during training from $\epsilon_{eval}$ that governs this deviation during evaluation. So the evaluation policy using the final estimated Q function $\hat{Q}^{(I)}$ becomes the solution of the optimization:

$$\pi_{eval}(\cdot|s) = \arg\max_{\pi \in \Delta^{|\mathcal{A}|}} \sum_{a \in \mathcal{A}} \hat{Q}^{(I)}(s,a)\pi(a|s), \quad s.t. \quad \sum_{a \in \mathcal{A}} \hat{u}(s,a)\left|\pi(a|s) - \hat{\beta}(a|s)\right| \leq \epsilon_{eval}. \quad (8)$$

As we will see in Section 4, this choice allows us to capture a richer tradeoff between different methods instead of simply interpolating between behavior cloning and greedy RL.

**Algorithmic variants.** We will consider two variants of the algorithm:

1. Standard deep-SPIBB where we set $\epsilon_{eval} = \epsilon_{train}$.
2. Generalized deep-SPIBB where we tune $\epsilon_{eval}$ independently of $\epsilon_{train}$.

These variants will be discussed in greater depth in the next section where we compare the tradeoffs made by $\epsilon_{train}$ and $\epsilon_{eval}$ with those made by the baseline offline RL algorithms introduced above.

## 4 Comparing algorithmic tradeoffs

Each algorithm introduced above comes with a hyperparameter that governs the tradeoff between acting greedily and restricting the learned policy to be safe (or two hyperparameters in the case of generalized deep-SPIBB). The key difference between the algorithms is how they choose to modulate this tradeoff and which points they choose at the extremes.

**Extremal hyperparameter settings.** To understand the tradeoff that each hyperparameter governs, it is useful to understand what happens at the extremal values. The results of this analysis are summarized in Table 2. There are a few key takeaways from this analysis:

- All algorithms capture greedy RL at one extreme of the hyperparameters.
- All algorithms except for pessimism capture a variant of BC at another extreme setting of the hyperparameters. Note that due to the greedy nature of the policies defined in BCQ and CQL, they cannot exactly represent BC for a stochastic behavior policy and instead choose the action that has maximum probability under the behavior (which we will denote as argmax BC).

Table 2: A summary of what each algorithm reduces to at extreme settings of their hyperparameters.

| Algorithm | Extreme 1 | Extreme 2 |
|---|---|---|
| BCQ | argmax BC at $\tau = 1$ | Greedy RL at $\tau = 0$ |
| CQL | argmax BC as $\alpha \to \infty$ | Greedy RL at $\alpha = 0$ |
| Pessimism | Minimal uncertainty as $\alpha \to \infty$ | Greedy RL at $\alpha = 0$ |
| Deep-SPIBB | BC at $\epsilon_{train} = \epsilon_{eval} = 0$ | Greedy RL as $\epsilon_{train} = \epsilon_{eval} \to \infty$ |
| Gen Deep-SPIBB | BC at $\epsilon_{eval} = 0$ | Greedy RL as $\epsilon_{train} = \epsilon_{eval} \to \infty$ |
|  | One-step RL at $\epsilon_{train} = 0$ |  |

- Only generalized deep-SPIBB captures one-step RL (Brandfonbrener et al., 2021; Gulcehre et al., 2021) that performs one step of policy improvement at another extreme setting of the hyperparameters.

**What happens to pessimism at the extreme.** It is worth delving deeper into what happens for the pessimism algorithm as $\alpha \to \infty$. Note that we can view the pessimistic algorithm as optimizing an augmented reward $\tilde{r}(s, a) = r(s, a) - \alpha \cdot u(s, a)$. As we send $\alpha \to \infty$ the impact of $r$ on $\tilde{r}$ tends to zero. As a result, we end up learning a policy that approaches $\pi_u(a|s) = \mathbb{1}[a = \arg\max_{a'} Q_u^*(s, a')]$ where $Q_u^*$ is the Q function of the optimal policy for the reward function $-u(s, a)$. We will call $\pi_u$ the minimal uncertainty policy. One key difference is that the minimal uncertainty policy can prefer policies that remain in states that are often observed in the dataset over actions chosen by the behavior. This difference is not necessarily bad in all cases, but provides a different inductive bias than the other algorithms considered.

The issues with pessimism become more problematic when the uncertainty function is poorly estimated (as it may be in high-dimensional state spaces). Since the default is to minimize the uncertainty, if the uncertainty estimate has some accidentally underestimated uncertainties there is no way to remedy the situation by tuning the hyperparameter $\alpha$. Sending $\alpha \to \infty$ will act greedily with respect to the negative uncertainty and thereby exploit the underestimated uncertainty estimate, yielding an unsafe policy. Alternatively, sending $\alpha \to 0$ will be greedy with respect to the estimated reward and yield a different unsafe policy. This is especially troublesome in realistic datasets where we expect the behavior to already give us a reasonable policy. In contrast, by defaulting to the behavior policy, deep-SPIBB can at least recover the performance of the behavior simply by tuning $\epsilon$, no matter the quality of the uncertainty estimates[2].

**The benefits of deep-SPIBB.** From this perspective, our deep-SPIBB algorithm has a few benefits. First, unlike pessimism, SPIBB is able to remain robust to poor quality uncertainty estimates by defaulting to the behavior policy. Second, unlike methods like BCQ and CQL that ignore state-based uncertainty, SPIBB can leverage uncertainty estimates when available.

Moreover, our generalized deep-SPIBB algorithm goes one step further by introducing a different default setting of the hyperparameters. When $\epsilon_{train}$ is set to 0, then the evaluation step will learn $Q^\beta$, as in one-step RL (Brandfonbrener et al., 2021; Gulcehre et al., 2021). Then as we vary $\epsilon_{eval}$ from 0 to infinity while fixing $\epsilon_{train} = 0$ we interpolate between BC and argmax of $\hat{Q}^\beta$, the greedy one-step policy. Capturing this algorithm as a special case (while the other baseline algorithms do not) provides further illustration that generalized deep-SPIBB is capturing a different tradeoff than the baseline algorithms.

## 5 EXPERIMENTS

We conduct an empirical analysis to evaluate how well deep-SPIBB incorporates uncertainty estimates and to compare deep-SPIBB to four baselines: BC, BCQ (Fujimoto et al., 2019b;a), pessimism

---

[2]Note that there may also be errors in the behavior estimate. However, here we are considering problems with large, high-dimensional state spaces, but finite action spaces. As a result, we expect it to be easier to solve the supervised learning problem of predicting action given state than to solve the uncertainty quantification problem of quantifying uncertainty over state and action, which implicitly requires learning a joint density model over $s$ and $a$ rather than just the model of $a$ conditioned on $s$

(Buckman et al., 2020; Jin et al., 2021), and CQL (Kumar et al., 2020). We run all of these algorithms along with deep-SPIBB on a diverse variety of datasets generated in the classic Cartpole and Catch environments as well as standard Atari benchmark datasets (Gulcehre et al., 2020). These experiments allow us to test how well deep-SPIBB is able to incorporate uncertainty across different domains from low-dimensional observation spaces (Cartpole) to image-based observations with simple dynamics (Catch) to image-based observations with complex dynamics (Atari). Our main finding is that deep-SPIBB consistently outperforms pessimism, suggesting that it does a better job of incorporating explicit uncertainty estimates. Our secondary finding is that when uncertainty is easier to estimate (as in Cartpole) deep-SPIBB substantially outperforms all the baselines, while in more challenging environments (as in Atari) it performs about the same as the strongest baseline method, CQL, suggesting a robustness to poor uncertainty estimates.

**Experimental setup.** Following CQL (Kumar et al., 2020), we build our deep-SPIBB algorithm and each of the baselines on top of QR-DQN (Dabney et al., 2018) using JAX (Bradbury et al., 2018) and the Acme framework (Hoffman et al., 2020). For all experiments, we tune each algorithm (BCQ, CQL, pessimism, and deep-SPIBB) across 4 values of the hyperparameters controlling the deviation from the behavior policy. All other training hyperparameters are held fixed. All algorithms have access to the same behavior and uncertainty estimates. Full details can be found in Appendix B.

## 5.1 BSUITE ENVIRONMENTS

**Datasets.** For our first set of experiments we consider two simple environments (cartpole and catch) from bsuite (Osband et al., 2019). In each environment we collect 5 different types of datasets with 10 seeds for each type of data (for a total of 50 datasets per environment). Cartpole has horizon and maximum return of 1000 and catch has horizon of 10 and returns bounded between -1 and 1. Datasets on cartpole have 20k transitions and datasets on catch have 2k transitions. The five dataset types in each environment are collected by as follows: (1) med is data collected by a DQN agent trained to medium performance (200 training episodes), (2) med_seed is a mixture of 5 different policies each trained to medium performance, (3) uni is a uniformly random policy, (4) uni_med is an equal mixture of data from a uniform policy and a medium policy, (5) uni_exp is an equal mixture of data from a uniform policy and an expert policy (trained for 500 episodes on cartpole, 1000 episodes on catch). We report mean and standard error across seeds. Results are shown in Figure 1.

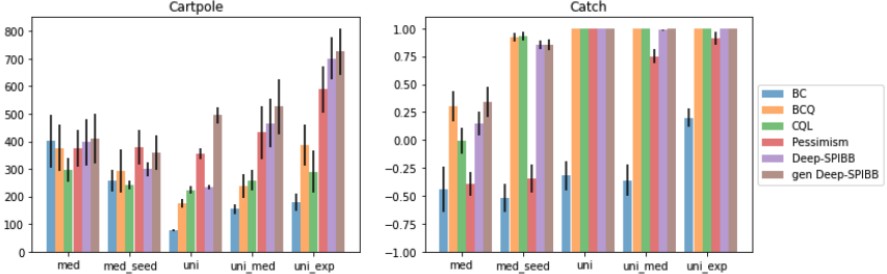

Figure 1: Final performance for OffRL agents trained on five dataset types across two environments. Error bars show standard error across ten seeds for dataset generation.

**Results.** These results show deep-SPIBB to be consistently the top performer across the suite of experiments and to emphasize our two main findings. First, consider the comparison to pessimism. Across all ten datasets, generalized deep-SPIBB outperforms pessimism and standard deep-SPIBB outperforms pessimism on nine out of ten datasets. The performance gap is particularly large on Catch, where the image-based inputs and smaller dataset size make uncertainty quantification more challenging. By defaulting to the behavior policy instead of the minimal uncertainty policy, deep-SPIBB is more robust to poor uncertainty estimates, while still being able to leverage good uncertainty estimates in Cartpole.

Second, consider the comparison to all baselines. Generalized deep-SPIBB is the top performer on nine out of ten datasets, with the only exception being a slight underperformance relative to BCQ and CQL on the Catch med_seed dataset. On Cartpole where the uncertainty estimation task is easier,

deep-SPIBB dramatically outperforms the uncertainty-free BCQ and CQL methods. On Catch, where uncertainty estimation is more difficult, deep-SPIBB does not outperform BCQ and CQL, but is able to match their performance while pessimism struggles due to the difficulty of uncertainty estimation.

## 5.2 ATARI ENVIRONMENTS

**Datasets.** Next we run deep-SPIBB and our four baselines on the atari 1% benchmark (Agarwal et al., 2020). Specifically, we use the data from RL Unplugged (Gulcehre et al., 2020) subsampled down to 1% of the trajectories for each run and report mean and standard deviation across three seeds for each environment. Results are shown in Figure 2.

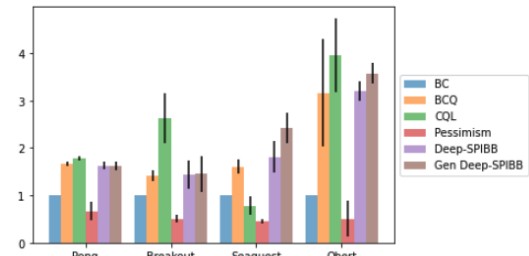

**Results.** Again these experiments back up our two main findings. First, compared to pessimism, deep-SPIBB is dramatically better. While deep-SPIBB consistently outperforms BC substantially, pessimism never even recovers the performance of BC. Uncertainty estimation in Atari is very difficult and thus pessimism's

Figure 2: Final performance for OffRL agents on atari 1% datasets. Results are normalized so that BC achieves a score of 1 and a randomly initialized policy achieves a score of 0. Error bars show standard deviation across three seeds for dataset generation.

choice to default to the minimal uncertainty policy can cause serious issues as seen here. In contrast, deep-SPIBB is much more robust to poor uncertainty estimates. Second, compared to all baselines, deep-SPIBB generally performs slightly better than BCQ and is competitive with CQL. Improving the performance of deep-SPIBB will likely require improved uncertainty quantification.

## 5.3 HYPERPARAMETER ABLATIONS

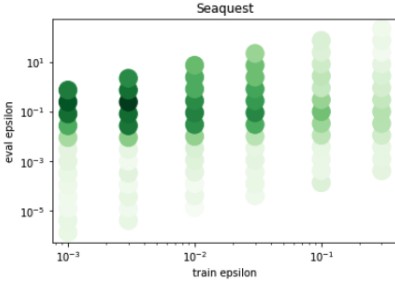 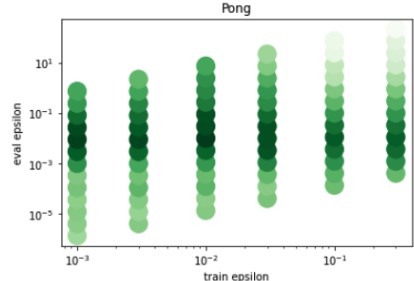

Figure 3: Sweeps across $\epsilon_{train}$ and $\epsilon_{eval}$ on Seaquest and Pong. Color indicates performance, darker is better. Generally, we find that lower $\epsilon_{train}$ and higher $\epsilon_{eval}$ is beneficial. This setting of hyperparameters places us closer to one-step RL.

To validate the usefulness of generalized deep-SPIBB over deep-SPIBB with $\epsilon_{train} = \epsilon_{eval}$, we conduct hyperparameter sweeps over the two different types of epsilon. Results are shown in Figure 3. Essentially, we find that it is often beneficial to set $\epsilon_{train}$ substantially lower than $\epsilon_{eval}$, especially on more challenging environments. This is consistent with the observations of Nadjahi et al. (2019); Brandfonbrener et al. (2021); Gulcehre et al. (2021) that the one-step algorithm that just performs one step of policy improvement is often a very strong algorithm (generalized soft-SPIBB captures the one-step algorithm for $\epsilon_{train} = 0$). Moreover, the more challenging environments and datasets likely yield worse uncertainty estimates, making it more risky to propagate values with low uncertainty since the uncertainty may be erroneously low. Thus, setting a lower $\epsilon_{train}$ can learn a more robust Q function, while allowing a larger $\epsilon_{eval}$ can yield improved performance.

The plots also show that the hyperparameters are somewhat independent in the sense that the optimal value of $\epsilon_{eval}$ is stable across different values of $\epsilon_{train}$. This observation can allow for more efficient hyperparameter tuning by avoiding a complete grid search.

### 5.4 UNCERTAINTY ABLATIONS

To validate our choices about how to parameterize the uncertainty estimates, we run deep-SPIBB wih several different uncertainty estimators. The most important decision in our uncertainty estimate is to use separate estimates of $u(s)$ and $\beta(a|s)$ to derive $u(s,a)$, as described above. This choice is validated by the results in Figure 4.

Essentially, we see major gains of our factored uncertatinty estimator over an uncertainty estimator trained on $s$ and $a$ jointly. Interestingly, just using the behavior component of the uncertainty estimator also outperforms the joint uncertianty estimator suggesting that the state-based part of the uncertainty is either not very important in this task or so poorly estimated in this high dimensional state space that it adds little to the performance.

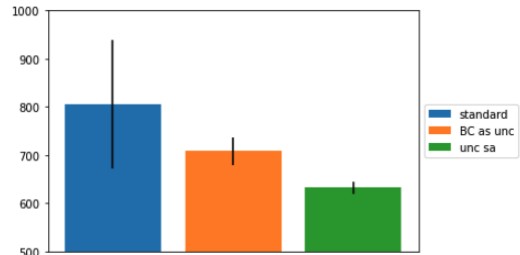

### 5.5 CONNECTION TO PRIOR RESULTS

This chunk of empirical analysis builds on the results already reported in the SPIBB papers Nad-

Figure 4: Uncertainty ablations on Seaquest. "BC as unc" uses $u(s,a) = \frac{1}{\sqrt{\hat{\beta}(a|s)}}$, and "unc sa" trains an uncertainty estimator that takes $s, a$ as input rather than using $u(s)$ combined with $\beta(a|s)$.

jahi et al. (2019); Simão et al. (2020) where the pessimistic algorithm RaMDP was found to perform as well as Soft-SPIBB under two conditions: (i) the uncertainty estimates are well-estimated, either from count-based statistical concentration bounds such as Hoeffding's inequality, or from handcrafted uncertainty measures based on states' similarity computed from their Euclidean distance in a well-adapted environment, and (ii) the intrinsic/extrinsic reward balance is fine tuned, meaning that the pessimism hyperparameter setting appeared as more sensitive than Soft-SPIBB's.

Our novel empirical results bring light on the robustness of these two approaches when the uncertainty estimator is more brittle. In Cartpole, with its low-dimensional observation space, we find that both are able to take a significant advantage over the uncertainty-free methods. In Catch where the state representation is more image-like, we see that pessimism's performance immediately crashes, while deep-SPIBB is more robust to these less-than-perfect uncertainty estimates and remains on-par with BCQ and CQL. Finally, in the Atari environments where the state is a complex image, pessimism is consistently and significantly worse than behavior cloning, which it cannot even fall back on.

This set of experiments with increasing difficulty in the uncertainty estimation (and the RL task) tells us that uncertainty-based algorithms are better than uncertainty-free algorithms when the uncertainty estimates are reliable but that out of the uncertainty-based algorithms, only deep-SPIBB is robust to bad estimates. While deep-SPIBB does not surpass CQL in hard environments for now, there is hope that advances in neural uncertainty estimation will allow it to do so in the future.

## 6 DISCUSSION

Here we have introduced the deep-SPIBB algorithm for inclorporating explicit uncertainty estimates into deep offine RL. We have seen that the deep-SPIBB mechanism of incorporating uncertainty into the policy improvement step is more performant and robust than the pessimism mechanism of incorporating uncertainty as a penalty in the evaluation step. When the uncertainty estimates are good, deep-SPIBB also improves over the uncertainty-free baselines of BCQ and CQL.

Our work does have a few limitations that are worth mentioning to inspire future work in these directions. First, it is not clear how to extend the deep-SPIBB algorithm to continuous action spaces. Second, like most algorithmic work in offline RL, deep-SPIBB has important hyperparameters ($\epsilon_{train}$ and $\epsilon_{eval}$) that govern the policy constraints. How to practically tune hyperparameters like these offline without interacting with the environment remains an open challenge, although recent work makes some progress (Paine et al., 2020; Zhang & Jiang, 2021). Finally, perhaps the most interesting direction for future work based on deep-SPIBB is to improve the uncertainty estimators. Our results suggest that access to better uncertainty estimators in challenging domains like Atari could dramatically improve deep-SPIBB over the uncertainty-free baselines.

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

## A  UNCERTAINTY ESTIMATOR

We take an approach based off of Bayesian ensembling with prior network from Ciosek et al. (2019). While that work focused on estimating uncertainty in supervised learning problems, we use the same technique with no labels to get a measure of state-based uncertainty. Explicitly, we create an ensemble of $B$ models that each take in a state $s$ and output an $M$-dimensional vector. Each component $f_i$ of the ensemble has a corresponding random prior function $p_i$ defined by a randomly initialized network. The $f_i$ networks are then trained to predict difference between the prior and some target (in our case, the target is gaussian noise $\epsilon_i$ sampled from $\mathcal{N}(0, \sigma_\epsilon)$ for each datapoint $i$). State-based uncertainty is then calculated as

$$\hat{u}(s) = \sqrt{\hat{\sigma}_\mu^2(s) + \alpha \hat{v}_\sigma(s)}, \tag{9}$$

$$\hat{\sigma}_\mu^2(s) = \frac{1}{MB} \sum_{i=1}^{B} \|f_i(s) - p_i(s)\|^2, \qquad \hat{v}_\sigma^2(s) = \frac{1}{B} \sum_{i=1}^{B} \left( \hat{\sigma}_\mu^2(s) - \frac{1}{M} \|f_i(s) - p_i(s)\|^2 \right)^2$$

## B  EXPERIMENTAL DETAILS

**Hyperparameters.**  First we will provide all of the hyperparameters used in the various steps of our training algorithms. Every algorithm is trained with the Adam optimizer.

Table 3: Hyperparameters for behavior estimation in cartpole and catch

| Hyperparameter | Value |
| --- | --- |
| Training steps | $1e4$ |
| Learning rate | $1e-3$ |
| Batch size | 256 |
| MLP width | 64 |
| MLP depth | 2 |
| Prior MLP depth | 1 |

Table 4: Hyperparameters for behavior estimation in Atari

| Hyperparameter | Value |
| --- | --- |
| Training steps | $1e5$ |
| Learning rate | $1e-4$ |
| Batch size | 256 |
| Network Architecture | DQN |

Table 5: Hyperparameters for uncertainty estimation in cartpole and catch

| Hyperparameter | Value |
| --- | --- |
| Training steps | $1e4$ |
| Learning rate | $1e-4$ |
| Batch size | 256 |
| MLP width | 256 |
| MLP depth | 2 |
| $M$ | 64 |
| $B$ | 5 |
| $\alpha$ | 1.0 |
| $\sigma_\epsilon$ | 0.1 |

Table 6: Hyperparameters for uncertainty estimation in Atari

| Hyperparameter | Value |
|---|---|
| Training steps | $1e5$ |
| Learning rate | $1e-4$ |
| Batch size | 256 |
| Network architecture | DQN |
| Prior network architecture | DQN - 1 dense layer |
| $M$ | 64 |
| $B$ | 5 |
| $\alpha$ | 1.0 |
| $\sigma_\epsilon$ | 0.1 |

Table 7: Shared hyperparameters for RL in cartpole and catch

| Hyperparameter | Value |
|---|---|
| Training steps | $1e5$ |
| Learning rate | $3-5$ |
| Batch size | 256 |
| Target update period | 1000 |
| MLP width | 256 |
| MLP depth | 2 |
| QR quantiles | 201 |
| Discount | 0.99 |

Table 8: Shared hyperparameters for RL in Atari

| Hyperparameter | Value |
|---|---|
| Training steps | $1e6$ |
| Learning rate | $3-5$ |
| Batch size | 256 |
| Target update period | 2500 |
| Network architecture | DQN |
| QR quantiles | 201 |
| Discount | 0.99 |

Each RL algorithm also has specific hyperparameters. For each algorithm we choose four values of the hyperparameter. We should note that for the deep-SPIBB and pessimism that rely on our learned uncertainty estimates, we normalize the values of $\epsilon$ and $\alpha$ respectively to the scale of the uncertainty estimator. We estimate the scale of the uncertainty function by just evaluating the mean of the uncertainty function on a batch of data from the training set.

Table 9: Algorithm-specific hyperparameters for cartpole and catch

| Algorithm | Hyperparameter | Value |
|---|---|---|
| BCQ | $\tau$ | [0.01, 0.03, 0.1, 0.3] |
| Pessimism | $\alpha$ | [0.3, 1.0, 3.0, 10.0] |
| CQL | $\alpha$ | [0.3, 1.0, 3.0, 10.0] |
| deep-SPIBB | $\epsilon_{train}$ | [0.1, 0.3, 1.0, 3.0] |
| gen deep-SPIBB | $\epsilon_{eval}$ | [0.001, 0.01, 0.1, 1.0] |

Table 10: Algorithm-specific hyperparameters for Atari

| Algorithm | Hyperparameter | Value |
|---|---|---|
| BCQ | $\tau$ | [0.01, 0.03, 0.1, 0.3] |
| Pessimism | $\alpha$ | [0.1, 1.0, 10.0, 100.0] |
| CQL | $\alpha$ | [0.3, 1.0, 3.0, 10.0] |
| deep-SPIBB | $\epsilon_{train}$ | [0.001, 0.01, 0.03, 0.1] |
| gen deep-SPIBB | $\epsilon_{eval}$ | [0.0001, 0.001, 0.01, 0.1] |

**Evaluation.** For evaluation we run 50 episodes of each trained RL model and take the mean. Plots in the text then report the mean and standard deviation of this mean value across training seeds. We report the results for the best performing hyperparameter out of those in the table for each algorithm.

**Compute.** All models are trained on various types of GPU on an internal cluster. Each run for cartpole/catch takes less than 15 minutes and each run on Atari takes less than 1 day.

**Asset licenses.** For completeness, we also report the licenses of the assets that we used in the paper: JAX Bradbury et al. (2018): Apache-2.0, Acme Hoffman et al. (2020): Apache-2.0, RL-unplugged Gulcehre et al. (2020): Apache-2.0, bsuite Osband et al. (2019): Apache-2.0.

