# OpenReview forum: "Incorporating Explicit Uncertainty Estimates into Deep Offline Reinforcement Learning"
_ICLR.cc/2023/Conference — Submitted to ICLR 2023_

### Official Review · Reviewer_uHxA · 2022-10-24

**Confidence:** 5
**Correctness:** 3
**Technical Novelty And Significance:** 1
**Empirical Novelty And Significance:** 1
**Recommendation:** 3

**Clarity, Quality, Novelty And Reproducibility:**

- How can you implement pessimism (Buckman et al., 2020; Jin et al., 2021) in your experimental setting? If you meant “pessimism” in (Buckman et al., 2020; Jin et al., 2021) by LCB, its ensemble-based implementation of LCB in fact has good empirical performance, see “PESSIMISTIC BOOTSTRAPPING FOR UNCERTAINTYDRIVEN OFFLINE REINFORCEMENT LEARNING”.

- SPIBB has never been explained and what it stands for.

- the claim “our proposed method is more performant and robust than the pessimism mechanism of incorportating uncertainty as a penalty in the evaluation step” is not useful your method is in some sense also pessimism mechanism as it constraints the learned policy to the behavior policy weighting by the state-action uncertain quantifier

- in the experiment section, you say that the purpose of the experiment is to test “how well deep-SPIBB incorporates uncertainty estimates” then “Our main finding is that deepSPIBB consistently outperforms pessimism, suggesting that it does a better job of incorporating explicit uncertainty estimates”. There is a questionable causal relationship between the performance of offline RL and incorporating uncertainty, that is, higher performance does not necessarily suggest better uncertainty corporation.

- the justification of pessimism at extreme hyperparameters is unfair and not useful. There is no good reason someone would set the uncertainty quantifier weight $\alpha$ to infinity in LCB. LCB would also implicitly implement behavior cloning with proper value of $\alpha$.

- ”Second, compared to all baselines, deep-SPIBB generally performs slightly better than BCQ and is competitive with CQL”. I don’t think it is competitive with CQL, Figure 2 shows CQL is clearly better than the proposed method.

- Figure 4 and its relevant claim are not clear. What is the y-axis of figure 4? how can we tell one uncertainty estimator is better than another by looking at Figure 4?

**Strength And Weaknesses:**

**Strength**

- Good classification of recent offline RL methods

- the paper is easy to read

**Weaknesses**

- Incremental methods with weak empirical results: the most interesting part is perhaps using the estimated behavior in the uncertainty quantifier. However, simple experiments in RL Unplugged cannot support the benefit of this modification.

- Justification and comparison with LCB is unfair: the extreme hyperparameter analysis is not useful and the paper missed literature of ensemble-based implementation of LCB. The paper also does not clearly explain how LCB is implemented in their baseline

- Significance of the work is low: Given the above reasons.

See my questions below for details.

**Summary Of The Paper:**

This paper proposes to enforce $\pi(a|s)$ close $\beta(a|s)$ to incorporate uncertainty at each state-action pair $(s,a)$ in enforcing $\pi(a|s)$ close the estimated behavior $\hat{\beta}(a|s)$ where the enforcement weight is controlled the uncertainty estimate $\hat{u}(s,a)$ (eq. 7). To compute uncertainty estimate $\hat{u}(s,a)$, they also incorporate estimated behavior policy as $\hat{u}(s,a) = \hat{u}(s) / \hat{\beta}(a|s)$ where $\hat{u}(s)$ is the ensemble-based estimator at state $s$. They claim that this proposed method is “more performant and robust than the pessimism mechanism [they meant LCB] of incorportating uncertainty as a penalty in the evaluation step”.

**Summary Of The Review:**

See the weaknesses and question sections.

---

### Official Review · Reviewer_hNm1 · 2022-10-25

**Confidence:** 3
**Correctness:** 4
**Technical Novelty And Significance:** 2
**Empirical Novelty And Significance:** 2
**Recommendation:** 3

**Clarity, Quality, Novelty And Reproducibility:**

**Clarity** the paper is clearly written

**Quality** the paper’s quality is somewhat below average

**Novelty** the paper’s novelty is limited due to prior works in similar area

**Reproducibility** detailed hyperparameters are provided in the appendix but no source code so not sure


**Strength And Weaknesses:**

### Strength
1. The proposed algorithm is very sensible and using uncertainty in offline RL seems very promising
2. The analysis of how different offline RL algorithms use uncertainty is very nice as it unifies different perspective
3. The paper overall is relatively easy to follow

### Weakness
1. The performance of the algorithm is not extremely compelling as it underperforms CQL in most settings.
2. The previous point would normally not be a big problem but the idea of using uncertainty for offline RL has already been proposed in the literature [1]. This work does not discuss [1] and furthermore, [1] seems to outperform CQL on a range of D4RL tasks. This casts doubt on the additive value of this work.
3. The ablation and analysis of different types of uncertainty could be improved. Currently, only two short paragraphs are dedicated to them, but it seems like uncertainty is extremely important for the proposed method and the proposed way of computing is new, so I believe more thorough analysis is warranted. For example, if we really believe that the quality of uncertainty estimation is the problem for the performance of the algorithm, then one could do an ablation on increasing the size of the ensemble and check if the performance improves.

**Reference**

[1] Why so pessimistic? Estimating uncertainties for offline RL through ensembles, and why their independence matters. Ghasemipour, et al.


**Summary Of The Paper:**

This paper proposes a new algorithm for offline RL,deep-SPIBB, that uses uncertainty estimation instead of pessimism for regularizing the behavior to prevent undesirable behaviors. The method uses an ensemble based uncertainty estimation on the state and MLE behavior policy estimate to estimate the uncertainty for a given state-action pair and using the uncertainty, the algorithm performs standard offline RL procedure with constraints that force the learned policy to not deviate from the MLE policy when there is high uncertainty. The algorithm also introduces different deviation thresholds for the policy improvement and policy evaluation step. Experimentally, the algorithm performs prior work on two simple environments and outperforms pessimism and is competitive with CQL on  Atari.

**Summary Of The Review:**

The paper is well written but is limited in significance and novelty. I am leaning towards reject.

---

### Official Review · Reviewer_eF2b · 2022-11-02

**Confidence:** 4
**Correctness:** 3
**Technical Novelty And Significance:** 2
**Empirical Novelty And Significance:** 2
**Recommendation:** 3

**Clarity, Quality, Novelty And Reproducibility:**

Clarity: Good, the writing is overall clear but could be improved, for examples

-  The abbr. SPIBB appeared without its full name in the main text.

- Page 5, "we use the approximation technique described in Nadjahi et al. (2019)." I would recommend including the optimization pseudo code in the appendix. Otherwise one has to go through Nadjahi et al. (2019) to locate it in their Appendix A.8 (Arxiv version).

- A lot of details are given in a quite verbal way, the readability could be improved.

Quality: Fair, some statements are not well supported and the experiments could be more comprehensive.

Novelty: Fair, novelty is limited because of Nadjahi et al. (2019) and  Burda et al. (2019). The discussion about constrained improvement step versus penalized evaluation step could be important. However, the attempts made in this version are not convincing enough.

Reproducibility: Good, code and hyper-params are provided.

**Strength And Weaknesses:**

Pros:

- The authors extend soft-SPIBB to large state-action space and the empirical performance seemed comparable to CQL.

- The design of the estimator $\hat{u}(s, a) := \hat{u}(s) / \sqrt{\hat{\beta}(a|s)}$ is sensible.

- The discussion about constrained improvement step versus penalized evaluation step is interesting.

Cons:

Limited novelty:

- The authors modified soft-SPIBB (Nadjahi et al., 2019)  by replacing (count-based) error functions $e(s, a)$ with neural uncertainty estimates $\hat{u}(s, a)$.

- Estimating state uncertainty via random priors is not new in reinforcement learning, e.g. RND (Burda et al., 2019).

Related works: prior works should be more explicitly explained

- RND is only mentioned in "there is a large literature from the deep learning community on uncertainty quantification that we can leverage for OffRL (... Burda et al., 2019; ...)", which does not emphasize that random priors methods have already been applied in reinforcement learning literature, although in an online setting.

- It is encouraged to briefly explain the essence of SPIBB and soft-SPIBB in e.g. preliminaries, so that it is easier to see which parts in Eqn (7) are proposed by the authors, especially for the audiences who are not familiar with soft-SPIBB.


Questions/additional comments:

- Introducing $\epsilon_{eval}$ seemed a bit odd to me. Do the authors mean: (a) first train $\hat{Q}$ until convergence with $\epsilon_{train}$ and (b) then do one-step policy improvement with $\epsilon_{test}$?

- Could the authors plot the training curves of experiments in 5.1 and 5.2? As Eqn (7) is now approximated by a greedy heuristic (if my understanding is correct), it is good to see whether it is requiring more iterations to converge.

- The comparison versus Pessimism is not necessarily fair. Deep-SPIBB keeps most essential parts of soft-SPIBB except using a neural quantifier instead of a count-based one. While Pessimism does not keep all key designs of e.g.  (Buckman et al., 2020; Jin et al., 2021), Pessimism should not recover theoretical guarantees of pessimistic approaches while using any count-based uncertainty quantifier because of its simplification. Therefore it does not fully support the claim made in the abstract, "we argue that the SPIBB mechanism for incorporating uncertainty is more robust and flexible than pessimistic approaches that incorporate the uncertainty as a value function penalty."

- I believe pessimistic approaches should be able to recover behavior cloning with proper choice of uncertainty penalty and also proper algorithmic designs. For example, see [1].

[1] Rashidinejad, Paria, et al. "Bridging offline reinforcement learning and imitation learning: A tale of pessimism." Advances in Neural Information Processing Systems 34 (2021): 11702-11716.

**Summary Of The Paper:**

This paper studies how to incorporate uncertainty estimations for offline reinforcement learning to prevent the learner from favoring regions of high uncertainty (which are often over-estimated). The authors extend safe policy improvement with soft baseline bootstrapping (soft-SPIBB) to large state-action space, where count-based uncertainty measures become infeasible, by leveraging a neural uncertainty estimator (using random priors).

**Summary Of The Review:**

Although the proposed algorithm is well-motivated (upon some existing works), its novelty is limited, and the discussions and experiments could use some improvements. I am leaning toward a (weak) reject at this moment.

---

### Decision · Program_Chairs · 2023-01-20

**Decision:**

Reject

**Justification For Why Not Higher Score:**

The novelty of the method is limited, and reviewers have pointed out a number of experimental studies that are required before the method can be thoroughly evaluated.  The superiority over the SOTA is also not clear.  No rebuttal was received to address these concerns.

**Justification For Why Not Lower Score:**

N.A.

**Metareview: Summary, Strengths And Weaknesses:**

This paper addresses the over-estimation issue in offline reinforcement learning via scalable and explicit uncertainty estimation, and the method is based on ensembles and MLE behavior policy estimate.  Promising experimental results are presented.

The paper is overall clearly written, and the estimator makes sense.  It also provides good literature review.

However, the novelty of the method is limited, and reviewers have pointed out a number of experimental studies that are required before the method can be thoroughly evaluated.  The superiority over the SOTA is also not clear.  No rebuttal was received to address these concerns.